Using a pacifier to decrease sudden infant death syndrome: an emergency department educational intervention

Walsh Paul 1 2 pfwalsh@ucdavis.edu
Vieth Teri 2
Rodriguez Carolina 2
Lona Nicole 2
Molina Rogelio 2
Habebo Emnet 2
Caldera Enrique 2
Garcia Cynthia 2
Veazey Gregory 2
1 University of California Davis, Department of Emergency Medicine , Sacramento, CA , United States
2 Kern Medical Center , Bakersfield, CA , United States
Zuo Li
Electronic publication date: 2014 Mar 13
Publication date: 2014
Volume: 2
Electronic Location ID: e309
Received 2013 Dec 28; Accepted 2014 Feb 25
Copyright: © 2014 Walsh et al.
Copyright year: 2014
Copyright holder: Walsh et al.
License: This is an open access article distributed under the terms of the Creative Commons Attribution License, which permits unrestricted use, distribution, and reproduction in any medium, provided the original author and source are credited.
License URL: https://creativecommons.org/licenses/by/3.0/

Keywords: Pacifier, Sudden infant death syndrome, Infant, Emergency department, Education in the emergency department

Funding: The Pediatric Emergency Medicine Research Foundation National Heart, Lung, and Blood Institute at the National Institutes for Health, the National Center for Advancing Translational Sciences, National Institutes of Health Award Number 5K12HL108964-02 & UL1 TR000002 This work was supported by The Pediatric Emergency Medicine Research Foundation, Long Beach, CA and by Award Number 5K12HL108964-02 from the National Heart, Lung, and Blood Institute at the National Institutes for Health, the National Center for Advancing Translational Sciences, National Institutes of Health, through grant number UL1 TR000002. The content is solely the responsibility of the authors and does not necessarily represent the official views of the National Heart, Lung, and Blood Institute or the National Institutes of Health or The Pediatric Emergency Medicine Research Foundation. The funders had no role in study design, data collection and analysis, decision to publish, or preparation of the manuscript.

==============================
Background. Pacifier use decreases the risk of sudden infant death syndrome (SIDS). An emergency department (ED) visit may provide an opportunistic ‘teachable moment’ for parents.

Objectives. To test the hypotheses (1) that caregivers were less familiar with the role of pacifiers in sudden infant death (SIDS) prevention than other recommendations, and (2) that an ED educational intervention would increase pacifier use in infants younger than six months, and (3) that otitis media would not occur more frequently in pacifier users.

Methods. We did an intervention-group-only longitudinal study in a county hospital ED. We measured pacifier use infants and baseline knowledge of SIDs prevention recommendations in caregivers. We followed up three months later to determine pacifier use, and 12 months later to determine episodes of otitis media.

Results. We analyzed data for 780 infants. Parents knew of advice against co-sleeping in 469/780 (60%), smoking in 660/776 (85%), and prone sleeping in 613/780 (79%). Only 268/777 (35%) knew the recommendation to offer a pacifier at bedtime. At enrollment 449/780 (58%) did not use a pacifier. Of 210/338 infants aged less than 6 months followed up 41/112 (37%) non-users had started using a pacifier at bedtime (NNT 3). Over the same period, 37/98 (38%) users had discontinued their pacifier. Otitis media did not differ between users and non-users at 12 months.

Conclusion. Caregiver knowledge of the role of pacifiers in SIDS prevention was less than for other recommendations. Our educational intervention appeared to increase pacifier use. Pacifier use was not associated with increased otitis media.

Introduction

In 2005 the American Academy of Pediatrics recommended that caregivers should offer infants between one and six months of age a pacifier (dummy, soother, binky) when putting them down to sleep. The recommendation was based on studies (De-Kun et al., 2005; Fleming et al., 1999; Arnestad, Andersen & Rognum, 1997; Fleming et al., 1996; Hauck et al., 2003) showing pacifier use is associated with decreased risk of sudden infant death syndrome (SIDS). Pacifier use also mitigates the SIDS risk associated with soft bedding and prone sleeping position (Task Force on Sudden Infant Death Syndrome, 2005). Our clinical experience suggested that relatively few parents knew the role of pacifiers in SIDS prevention; on the contrary they feared pacifiers increased ear infections or dental problems.

Alcohol abuse and injury prevention research suggest that emergency department (ED) visits represent ‘teachable moments’ during which educational interventions may be disproportionately effective (Johnson et al., 2007; Williams et al., 2005). In this case, a potentially effective intervention in the ED could be to recommend offering a pacifier at bedtime. Although a randomized controlled trial would be ideal, we felt it unethical to randomize some infants’ parents to receive knowledge that could prevent SIDS while withholding it from others.

Instead, we performed an intervention only trial in which we associated new pacifier use with (but could not attribute it to) our intervention and controlled for other variables. We conceptualized the conversion of non-user to user as a combination of infant and family factors, our intervention, overall community knowledge (which could vary with time), and knowledge dissemination as a direct result of our intervention.

We hypothesized (1) that infant caregivers were less familiar with the role of pacifiers in SIDS prevention than other recommendations, (2) that an ED educational intervention would increase pacifier use in infants aged between one and six months, and (3) that otitis media would not occur more frequently in infants using pacifiers.

Methods

Objectives

Primary objective

To compare self-reported primary caregiver knowledge of the recommendation that pacifier use is associated with decreased SIDS risk with self-reported primary caregiver knowledge of the recommendations that infants should sleep on their backs.

Secondary objectives

To compare knowledge of the pacifier recommendation with other SIDS prevention recommendations; namely that infants should not co sleep with parents, should not have additional blankets or toys in the crib, and that baby should not be exposed to secondhand tobacco smoke.

To determine what proportion of primary caregivers who received the intervention introduced a pacifier if their infant was aged one to six months and not already using one.

To determine if the occurrence of otitis media, or recurrent otitis media differed between pacifier users and nonusers during the 12 months following enrollment.

To identify possible associations between caregiver and infant characteristics and knowledge of the pacifier recommendation.

To identify possible associations between caregiver and infant characteristics and conversion from pacifier nonuser to user and vice-versa.

Design

We conducted a longitudinal study of an educational intervention between 11/26/2008 and 8/1/2011 including a 12 month period of follow up without additional patient accrual.

Setting

The study site was a teaching county hospital with emergency medicine, family practice and OBGYN residencies serving a mixed rural, suburban and urban population.

Subjects

Inclusion criteria

All infants younger than 12 months of age and their primary caregivers were eligible. Research assistants (RA) worked four or eight hour shifts including nights, weekends and holidays. Because of potentially non-random gaps in RA coverage we considered our sampling to be convenience rather than consecutive.

Exclusion criteria

Subjects were excluded for refusal of consent, being in foster care or custody.

Study definitions

Pacifier use was defined as parentally reported use of the pacifier when the infant was being put down to sleep. Knowledge of a particular recommendation was defined as the primary caregiver affirming that they were aware of the recommendation when specifically asked. When parents were uncertain whether or not they knew about a recommendation the answer was classified as missing. We defined early pacifier discontinuation as discontinuation of pacifier use before six months of age in an infant who was a pacifier user at enrollment. Otitis media was defined as parental report of a health care provider diagnosing otitis media. Recurrent otitis media was defined as three or more such episodes.

Outcome measures

Our primary outcome measure was the percentage of primary caregivers who reported knowing that offering infants between one and six months of age a pacifier at bedtime decreases the risk of SIDS. Our secondary outcome measures included the percentage of primary caregivers with knowledge of the other SIDS prevention recommendations, the percentage of caregivers who started offering their infant a pacifier (if that infant was aged one to six months and was not already a pacifier user), and parentally reported occurrences of otitis media at 12 months following enrollment. Our outcome measures for associations between caregiver and infant characteristics and knowledge of the pacifier recommendation and conversion from pacifier nonuser to user were odds ratios (OR).

Intervention

The survey and educational intervention were administered by an RA or investigator. The initial survey inquired about pacifier use, parental knowledge of SIDS prevention strategies, household characteristics, and other demographic characteristics. The key SIDS-prevention strategies inquired after were: (a) infants should not sleep in the same bed as their parents; (b) infants should sleep on their back, not prone; (c) stuffed toys, comforters, blankets etc. should not be in the crib; (d) parents should not smoke; and (e) infants between one and six months of age should be offered a pacifier when being put down to sleep. This baseline survey was conducted face to face with the primary caregiver and is included as Appendix S1. Prior to implementation, we tested pilot versions to reduce ambiguity.

The educational intervention consisted of the RA discussing SIDS prevention with the parents by explaining the contents of a printed color one-page brochure. Spanish speakers or a telephone translation service was used for Spanish speaking caregivers. This brochure is shown in Appendix S2. After the intervention, the brochure was given to the parents. RAs were trained with two hours of didactic lectures about SIDS and SIDS prevention. They were also trained in study enrollment procedures using scripts, role play sessions, and by observation of a study investigator. RA training was repeated and reinforced regularly during the study.

Follow-up telephone calls were made at three and 12 months after enrollment. Five attempts at telephone contact were made on different days and at different times for each subject at each time point. If follow-up failed we checked the coroner’s records for vital status. Spanish and English speaking RAs made the follow-up calls. During the three-month follow-up telephone-call, we questioned caregivers about pacifier use and about how many other people they had told of the intervention. At 12-months follow-up, we inquired about the number of episodes of otitis media with which the child had been diagnosed.

The initial survey was collected with pen and paper and keyed into a customized Filemaker-pro database (Filemaker Inc, Santa Clara, CA) by RAs. The investigators reviewed all cases for data entry errors. Follow-up data was entered directly into the database during the telephone call.

Statistical methods

Primary outcome

We compared knowledge of the pacifier recommendation with the back to sleep and other SIDS prevention recommendations using Fisher’s exact test.

Secondary outcomes

We attempted to identify infant, caregiver and household characteristics factors associated with pacifier use. We performed univariate analysis of these variables: pacifier use initiated in the hospital, NICU stay, primary care provider, primary caregiver, self-described primary caregiver race/ethnicity, caregiver age, recollection of prior clinic and discharge education, medical insurance type, number of bedrooms, number of siblings, and their interactions. Variables with a p value of <0.20 were considered for multivariable modeling. We ultimately retained only those with a p-value of ≤0.05. We used the same approach to identifying factors associated with knowledge of the pacifier and other recommendations.

We expected that conversion from pacifier non-user to user between one and six months of age would be a function of (a) infant, caregiver and household characteristics (listed above), (b) overall community knowledge of the role of pacifiers which could vary with time (a secular trend), (c) knowledge spread among caregivers as a direct result of the intervention and (d) our intervention. We could not directly measure the effect of our intervention without withholding it. We therefore adjusted for the other factors in the same manner as when attempting to identify and associated the remaining effect with our intervention.

Secular trends in overall community knowledge

We addressed community knowledge by measuring awareness of the role of pacifiers in SIDS prevention in infants up to 12 months of age and comparing the baseline knowledge of newly enrolled caregivers in each four month period of recruitment for each recommendation.

Physical proximity

We addressed physical proximity by creating two proximity measures defined as the number of prior participants living within five and 20 min driving time. Times were calculated using Google maps (Ozimek & Miles, 2011).

Social proximity

We addressed the effect of social proximity, as a proxy for social proximity itself, by asking caregivers at follow-up with how many other people had shared the pacifier recommendations. We included the number of people prior subjects reported telling as a variable in our model.

SIDS by definition occurs up to 12 months of age. We included infants up to 12 months of age because we were assessing knowledge of recommendations to prevent SIDS. Because the pacifier recommendation applies only up to six months of age, we limited intervention effect estimates to infants aged less than six months. We included all infants when estimating the effect of pacifier use on the incidence of otitis media.

We calculated the efficacy of the intervention as the proportion of non-users who became users and who were younger than six months old at first follow up. We performed an intention to treat analysis for those successfully followed up but who, because of delays in successfully contacting the caregiver, were actually over six months of age at follow up and a treatment received analysis including those only who were actually under 6 months of age at this follow-up. We used published estimates of the numbers needed to treat (pacifier use) (NNT) to prevent one death from SIDS to estimate of the number of infants whose caregivers would need to be educated to prevent one death from SIDS (Hauck et al., 2003).

We compared the prevalence of parent-reported diagnoses of otitis media, and recurrent otitis media (defined as three or more episodes) between pacifier users and never-users with Fisher’s exact test. Some have argued that pacifier use increases the risk for otitis media (Uhari, Mäntysaari & Niemelä, 1996). Our clinical experience also suggested that this was a concern for some parents. Therefore we felt it important to collect data on otitis media despite the fact that it tends to be more common in children over two years of age than in infants.

We compared the characteristics between those in whom follow up was successful and those in whom it was not. We performed post hoc exploratory analysis of factors associated with conversion from pacifier user to nonuser using univariate analysis and logistic regression.

We managed and analyzed study data using Stata 12 (Statacorp LLP, College Station TX). Kern Medical Center’s institutional review board approved the study. Written informed consent was obtained from the available adult with nearest next of kin.

Results

We enrolled 799 infants. Nineteen patients were excluded for repeated enrollment. One infant died of SIDS and one of pneumonia. Both were pacifier users at baseline. The primary caregiver was usually the mother or grandmother. The median age of mothers (who were sole caregivers) was 24 (IQR 10); grandmothers were in their 40s–50s (median and mean 50 IQR14). Sample characteristics and baseline pacifier use are detailed in Table 1. Patient flow through the study is shown in Fig. 1.

Figure 1 Patient flow through the study.

Table 1 Description of infants, caregivers and their households.

Medi-Cal is Medicaid in California.

		Number	(%)	Median	IQR	
Age	Total sample (in months)	780		3.90	6.28	
Primary caregiver					
	Mother alone	608	78			
	Mother & Father	29	4			
	Grandmother alone	15	2			
	Grandmother & Mother	90	11			
	Other	38	5			
	Father alone	0	0			
Insurance					
	Medi-Cal	656	84			
	Private	22	3			
	Uninsured	32	4			
	Declined to answer	70	9			
Pacifier use					
(Younger than 6 months at enrollment)	509	65			
	Uses Pacifier when sleeping					
	Never	271	53			
	Sometimes	166	33			
	Usually	30	6			
	Always	42	8			

Baseline knowledge of SIDS prevention recommendations

Caregiver knowledge of recommendations is in Table 2. Pacifier use was the least well known recommendation, 268/777 (34%), compared with 613/780 (79%) for supine sleeping (p < 0.001). Pacifier was also significantly less well known than any other recommendation (p < 0.001). African–Americans had consistently poorer baseline knowledge of the recommendations but made up only 8% of the sample. Knowledge of one recommendation was associated with knowledge of the pacifier in univariate analysis. This effect was smaller for advice against smoking which appeared to be known to parents regardless of knowledge of other SIDS prevention strategies. All of these effects were weaker in multivariable analysis. These are shown in Appendices S3 and S4.

Table 2 Baseline knowledge of SIDS prevention recommendations by primary caregivers.

Recommendation	Overall	First child	≥3 children	Carer ≤ 35y	Carer > 35y	
	n	(%)	n	(%)	n	(%)	n	(%)	n	(%)	
Not to sleep in same bed as an adult	469/780	60	116/185	63	124/214	58	339/660	51	80/119	67	
Infant to sleep on his back	613/780	79	143/185	77	166/214	78	518/661	78	95/119	80	
No blankets, stuffed toys	589/776	76	136/184	74	158/213	74	500/657	76	89/119	75	
Caregivers should not smoke	660/776	85	166/183	91	178/213	84	566/657	86	96/119	81	
Offer infant pacifier to sleep	268/777	34	64/184	35	76/213	36	223/658	34	45/119	38	

Pacifier use

At baseline 331/780 (42%) used a pacifier. Among infants aged less than three and 3–6 months pacifier use was 166/338 (49%) and 71/171 (42%) respectively. Pacifier use was more frequent among younger infants of younger mothers and among those who been given a pacifier in the hospital.

The initiation of pacifier use in the newborn nursery and parental knowledge that pacifiers decrease SIDS were the strongest predictors of pacifier use at enrollment. This suggests that initiating pacifier use in the newborn nursery and telling parents that pacifiers decrease SIDS at that time could be an effective strategy. Increasing infant age decreased the odds of pacifier use (9% per month of life). Older caregivers were also less likely to offer a pacifier (odds decrease 3% for each additional year of age or 26% for each additional 10 years of age). These associations with baseline pacifier use are shown in Table 3. Parents also indicated that advice given personally by a physician was highly influential.

Table 3 Factors associated with pacifier use at enrollment.

Ratio of bedrooms to children includes primary caregiver’s bedroom.

Variable	Odds ratio (95% CI)	
Infants’ age (per month)	0.91 (95% CI [0.87–0.95])**	
Caregivers’ age (per year)	0.97 (95% CI [0.95–0.99])**	
Ratio of bedrooms to children	1.28 (95% CI [1.08–1.53])**	
Given pacifier in the hospital	1.75 (95% CI [1.28–2.40])**	
Knew pacifier recommendation	1.39 (95% CI [1.01–1.92])*	
Notes.

* p < 0.05.

** p < 0.01.

CI confidence interval

Table 4 Comparison of characteristics of those in whom 3 month follow up was successful and those in whom it failed.

SIDS counseling at discharge; refers to discharge after child birth.

Factor	Level	Follow up failed	Followed up	p	
N		284	496		
Male		161 (56.7%)	270 (54.4%)	0.54	
Age, median (IQR)	months	3.9 (1.2, 7.0)	3.91 (1.2, 8.0)	0.29	
Method of feeding at enrollment	bottle	163 (65.7%)	302 (61.8%)	0.66	
	bottle and solids	1 (0.4%)	2 (0.4%)		
	breast	29 (11.7%)	53 (10.8%)		
	breast and bottle	55 (22.2%)	131 (26.8%)		
	solids only	0 (0.0%)	1 (0.2%)		
Had NICU stay		48 (19.3%)	106 (21.5%)	0.47	
Number of bedrooms, median (IQR)		2.5 (2, 3)	3 (2, 3)	0.20	
Number of siblings, median (IQR)		1 (1, 3)	2 (1, 3)	0.37	
Pacifier started in the hospital	No	155 (55.0%)	260 (52.6%)	0.53	
	Yes	127 (45.0%)	234 (47.4%)		
Caregiver age, median (IQR)	years	23 (20, 30)	26 (21, 31)	0.011	
SIDS counseling at discharge	No	134 (47.2%)	272 (54.8%)	<0.001	
	Yes	111 (39.1%)	206 (41.5%)		
	Missing	39 (13.7%)	18 (3.6%)		
SIDS counseling at well baby clinic	No	150 (52.8%)	354 (71.3%)	0.001	
	Yes	97 (34.4%)	134 (27.0%)		
	Missing	37 (13.0%)	8 (2%)		
Pacifier user at enrollment	No	155 (54.6%)	294 (59.3%)	0.200	
	Yes	129 (45.4%)	202 (40.7%)		
Notes.

NICU neonatal intensive

SIDS sudden infant death syndrome

IQR interquartile range

Missing declined to answer, could not recall or unsure

Effect of the intervention

We completed three-month follow-up in 496/780 (64%) patients. The characteristics of those in whom three month follow up failed and was successful are described in Table 4. Those who failed follow up had caregivers who were slightly older and who were less likely to recall being counseled in SIDS prevention strategies either at discharge following birth or in the clinic. These effects disappeared in multivariable analysis. Twelve month follow-up was successful in 391/780 (50%) infants. Follow up tended to be less successful in younger children and with older parents but these effects disappeared in multivariable analysis. Overall pacifier use at three-month follow up was 192/496 (39%); this comprised younger infants starting and older infants discontinuing pacifier use (Fig. 2).

Figure 2 Pacifier uptake and discontinuance rates by age group.

Table 5 Comparison of pacifier users who did and did not discontinue use before six months of age.

SIDS counseling at discharge; refers to discharge after child birth.

Factor		Did not stop	Stopped using pacifier	p	
N		61	37		
Male		30 (49%)	27 (73%)	0.021	
Age, median (IQR)	months	1.02 (0.56, 1.69)	1.32 (0.93, 1.92)	0.340	
Age at follow up (months), median (IQR)	months	4.69 (4.10, 5.75)	5.95 (4.63, 6.71)	0.008	
Method of feeding at enrollment	bottle	32 (53%)	20 (54%)	0.980	
	breast	9 (15%)	5 (14%)		
	breast and bottle	19 (32%)	12 (32%)		
Had a NICU stay		16 (26%)	6 (16%)	0.280	
Number of bedrooms, median (IQR)		3 (2, 3)	3 (2, 3)	0.750	
Number of siblings, median (IQR)		2 (0, 3)	1 (1, 3)	0.930	
Pacifier initiated in hospital	No	29 (48%)	26 (70%)	0.028	
	Yes	32 (52%)	11 (30%)		
Caregiverage, median (IQR)	Years	24 (20, 30)	22 (20, 28)	0.480	
SIDS counseling at discharge	No	37 (61%)	24 (65%)	0.910	
	Yes	22 (36%)	12 (32%)		
	Missing	2 (3%)	1 (3%)		
SIDS counseling at wellbaby clinic	No	45 (74%)	29 (78%)	0.370	
	Yes	15 (25%)	6 (16%)		
	Missing	1 (2%)	1 (3%)		
Number of bedrooms, median (IQR)		3 (2, 3)	3 (2, 3)	0.750	
Fed by bottle only (at enrollment)	No	28 (47%)	17 (46%)	0.940	
	Yes	32 (53%)	20 (54%)		
Notes.

NICU neonatal intensive

SIDS sudden infant death syndrome

IQR interquartile range

Missing declined to answer, unsure or could not recall

Intention to treat analysis

Three month follow-up was successful in 210/338 (62%) infants who were aged less than three months at enrollment (i.e., aged less than 6 months at follow-up). We contacted 112/172 (65%) of previous non users and 98/166 (59%) baseline pacifier users. Of the nonusers 41 (37%) had started using a pacifier at bedtime. Over the same time period, 37/98 (38%) users younger than three months had discontinued their pacifier. Characteristics of infants who did and did not discontinue their pacifiers are in Table 5.

Treatment received analysis

Sixty-two infants who were expected to be less than six months of age at three-month follow up were in fact older than six months because of delays in successfully completing follow up. Excluding these infants at three-month follow up; 70/148 (47%) were pacifier users at enrollment. Following the intervention 33/78 (42%) of nonusers has started using a pacifier and 20/70 (29%) had discontinued pacifier use.

Assuming that none of the nonusers would have spontaneously become users without the intervention, and ignoring any reduction in early pacifier discontinuation attributable to the intervention, we estimate the number of non-pacifier-using infants needed to treat (educate caregiver for infants aged <3 months) to gain an additional user is 3, (95% CI [2, 4]). The NNT for pacifier use to prevent one case of SIDS was 2733, yielding an NNT (educational intervention) to prevent one SIDS case of 8199 (Hauck et al., 2003).

Only infant age was significantly associated with starting pacifier use after the intervention OR 0.77 (95% CI [0.63, 0.94]) i.e., the odds of adoption of a pacifier decreased by 23% for each additional month of age. Participants reported telling a total of 947 (median 1, IQR 3), other individuals about the pacifier recommendation. The number of people told by prior participants did not affect any outcome. We were able to calculate physical proximity for 281,250 participant dyads. We found no association between physical distance between a subject and prior participants and pacifier adoption. We found no effect for secular trend measured in four month intervals. None of the other infant, caregiver or household variables tested had any impact on pacifier adoption.

Increasing age was not associated with change in sleeping position but did decrease pacifier uptake.

Factors associated with discontinuing pacifier use were infant age, male gender and not initiating pacifier use in the hospital (Table 6). Because of the small numbers in this subgroup only two of these three simultaneously maintained a p value of less than 0.05 in multivariable analysis. The models however had similar characteristics and are reported in Table 6. Interestingly, starting a pacifier in the hospital prior to discharge was associated with both with a higher rate of pacifier use, (OR 1.75) and less early discontinuation of the pacifier (OR 0.36). This does suggest that a simple intervention, namely starting pacifiers in the hospital would increase pacifier use. While this may not hinder breastfeeding in motivated parents, this may not be the case for less motivated mothers and so this decision may need to be individualized.

Table 6 Multivariable models of factors associated with early discontinuation of pacifier.

The differences between these models is slight.

Factor	OR
(95% CI)	p	OR
(95% CI)	p	OR
(95% CI)	p	
Pacifier started in
hospital	0.43
(0.16, 1.10)	0.079	0.362
(0.15, 0.90)	0.028			
Male	2.50
(0.97, 6.43)	0.058			2.96
(1.17, 7.44)	0.021	
Age at follow up	1.28
(1.04, 1.58)	0.019	1.27
(1.04, 1.55)	0.017	1.29
(1.05, 1.60)	0.002	
Model							
Pseudo R2	0.1228	0.002	0.094	0.002	0.098	0.002	
AIC	1.250		1.262	1.262	1.256		
BIC	−316.507		−317.854		−318.441		
Notes.

OR odds ratio

AIC akaike information criteria

BIC Bayesian information criteria

At one year follow up parents reported 156 episodes of otitis media in 391 infants (40%) with 35 (9%) having three or more infections and eight parents reporting six or more episodes. Overall, the prevalence of parent-reported otitis media in ‘never-users’ was indistinguishable from pacifier users. (Fisher’s exact test p = 0.808). Among never-users there were 18/156 (12%) infants with recurrent (three or more episodes of) otitis media compared with 17/112 (15%) among any time pacifier users (Fisher’s exact test p = 0.463). Multivariable regression of the number of episodes similarly showed no significant relationships between any variable tested.

Discussion

Parental knowledge of the role of pacifiers in SIDS reduction was much less than for other SIDS prevention recommendations. Our educational intervention appeared to increase pacifier use. We did not see an association between parent-reported otitis media and pacifier use.

Our subjects’ knowledge of other parenting practices known to reduce SIDS was at least as good as that reported elsewhere among professional child minders and parents (Moon & Oden, 2003; Moon et al., 2010). Perhaps it is unsurprising that parental knowledge of the role of pacifiers was less than for the other recommendations; the same is true of health care providers (Moon et al., 2007; Eron et al., 2011). The better known recommendations substantially predate the pacifier use recommendation. Unlike recommendations against smoking and non-supine sleeping positions; the pacifier recommendation has been controversial (Fleming, Blair & McKenna, 2006). Recommendations to use a pacifier compete with some mothers’ and dentists’ fears that pacifiers will harm infants’ developing mandibles, (Pansy et al., 2008; Vazquez-Nava et al., 2006; Warren et al., 2005) impair breastfeeding, (Howard et al., 2003; Scott et al., 2006) or increase otitis media (Uhari, Mäntysaari & Niemelä, 1996).

Educational interventions addressing bed sharing, smoking, and sleeping position have been shown to be effective in changing parental behavior, (Rasinski et al., 2003; Gibson et al., 2000; Moon, Oden & Grady, 2004) and epidemiological studies show community-wide education decreases SIDS deaths (Davidson-Rada, Caldis & Tonkin, 1995; Kiechl-Kohlendorfer et al., 2001). The effect size of our intervention on pacifier use was comparable to that of other educational interventions designed to decrease prone sleeping and bed sharing (Moon, Oden & Grady, 2004). This is encouraging because the change in behavior occurred in the absence of the expansive multi-pronged approach of other successful SIDS prevention interventions (Davidson-Rada, Caldis & Tonkin, 1995). Moreover the marginal cost of healthcare providers educating parents during their ED visit is low and is feasible in any setting.

Implementing an intervention such as ours is not a trivial exercise. The very high NNT to prevent one SIDS case reflects the rarity of SIDS. Consistent with studies of the ‘Back to Sleep’ campaign, we found during our discussions with parents that they especially value physicians’ advice. Further study to determine the efficacy and costs of an opportunistic targeted approach by emergency physicians is warranted (Willinger et al., 2000). We speculate that individual emergency physicians discussing SIDS prevention strategies, and specifically discussing the role for pacifiers during history taking, would incur minimal marginal cost and would be much more efficient. This does not detract from the role of primary care providers in instructing parents regarding SIDS prevention strategies.

Limitations

This study has limitations. We assumed new pacifier use was a result of our intervention and discounted delayed pacifier discontinuance that might have resulted from our intervention. This assumes that parents would not discontinue pacifier use as a result of our intervention. We feel that this is a reasonable assumption. The ideal approach would be to randomize parents to receive this SIDS risk reduction information or not. We felt that withholding information that is known to decrease SIDS to measure the effectiveness of our intervention would be unethical. Even if we were to accept that not providing this information amounted to ‘usual care’ we would still have had to obtain consent from ‘usual care’ patients in order to perform the initial and follow-up surveys. Parents might well inquire why pacifier use was being asked after. They would likely be unimpressed if they were being assigned to ‘usual care’, and that we intended to withhold lesser known but consistently effective pacifier recommendations to prevent SIDS. It is difficult to conceive how a study design that willfully withheld such information from parents would not damage the trust between researchers and their community. This would be particularly the case should a SIDS case occur in a non-pacifier using infant in the control group. We addressed this limitation as best we could by controlling for other factors that may affect caregiver knowledge.

Our analysis of factors associated with adoption and early discontinuation of pacifier was limited by small numbers despite our large overall sample size. Nonetheless our findings that older age decreases pacifier uptake and increases pacifier discontinuation seems reasonable. Moreover we also identified a modifiable risk factor for early discontinuation which was the same as for actual use suggesting internal consistency in our findings. We accepted parental reports as being accurate. However we had no way to verify the veracity of their statements or of actual pacifier use or actual otitis media. We also did not quantify beyond “yes” or “no” for knowledge of each SIDS prevention recommendation. These would however tend to bias our comparisons of caregiver knowledge of different recommendations to the null.

We standardized the time to follow-up rather than choosing to follow-up when the child was aged six months. This facilitated assessment of the intervention but was less patient centered. We relied on caregiver reporting and recall of outcomes. Because we used only a single site, external validity is unproven. Nonetheless our finding that parental knowledge of other SIDS recommendations was similar to that reported by other investigators supports the external validity of our findings (Gibson et al., 2000). Our study was powered to demonstrate increased pacifier adoption not a decreased SIDS rate. We did not measure changes in the adoption of other recommendations. Collecting this additional comparative data would have lengthened the interview process and potentially decreased caregiver cooperation. We also had difficulty completing follow-up, a common difficulty in patient populations such as ours. There were some differences between those in whom follow up succeeded and failed. Caregivers in whom follow up failed were less likely to answer questions about SIDS prevention education in either the clinic or the hospital; they were also slightly older. However these differences did not persist in multivariable analysis. Finally, we had to rely on a simple pamphlet without the benefit of language optimization which could have increased efficacy (Buller et al., 2000).

Conclusion

Parental knowledge of the role of pacifiers in SIDS prevention was modest and much less than for other recommendations. Starting a pacifier prior to hospital discharge after birth was associated with greater use and lower discontinuation rates in the following year. Our broadly targeted ED-based educational intervention was labor intensive but appeared successful in increasing pacifier use. Pacifier use was not associated with increased otitis media.

Supplemental information

Appendix S1 Survey instrument as it appears to research assistants doing data entry

Click here for additional data file.

Appendix S2 Informational brochure used during our educational intervention and given to the primary caregiver to keep

Click here for additional data file.

Appendix S3 Univariate analysis of factors associated with primary caregiver knowledge of each of five pacifier prevention recommendations

Click here for additional data file.

Appendix S4 Multivariable analysis of the factors from Appendix S3

Click here for additional data file.

The authors gratefully acknowledge the assistance of Sheriff’s Deputy Dawn Ratliff Supervising Deputy Coroner for Kern County for her interest in SIDS prevention and her assistance with this project, Stephen J. Rothenberg for his statistical advice and Nathan Kuppermann for his comments on the manuscript.

Additional Information and Declarations

Competing Interests

Author Contributions

Human Ethics

Paul Walsh, Teri Vieth, Carolina Rodriguez, Nicole Lona, Rogelio Molina, Gregory Veazey, Emnet Habebo, Enrique Caldera, Cynthia Garcia are or were employees of or research volunteers at Kern Medical Center.

Paul Walsh conceived and designed the experiments, performed the experiments, analyzed the data, contributed reagents/materials/analysis tools, wrote the paper, prepared figures and/or tables and reviewed drafts of the paper.

Teri Vieth conceived and designed the experiments, performed the experiments, analyzed the data and reviewed drafts of the paper.

Carolina Rodriguez, Nicole Lona, Rogelio Molina, Emnet Habebo, Enrique Caldera, Cynthia Garcia and Gregory Veazey performed the experiments and reviewed drafts of the paper.

The following information was supplied relating to ethical approvals (i.e., approving body and any reference numbers):

Kern Medical Center IRB: Approval 0826.

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
