# Peer review of "Using a pacifier to decrease sudden infant death syndrome: an emergency department educational intervention"

_PeerJ, doi:10.7717/peerj.309_

## Round 0.1 · original submission · Major Revisions

Thank you for submitting your paper to PeerJ. Your manuscript has been reviewed by expert reviewers, who have expressed some enthusiasm for your review. However, the reviewers have identified a number of concerns about your manuscript, which are outlined in the full text of the reviewer report, appended at the end of this email. In the event that you would like to address these concerns, I am returning your manuscript to you for MAJOR REVISION. Accordingly, please provide a pt-by-pt response letter along with your revision.

Reviewer 1 ·

Basic reporting

1. It would be better if one flow chart can be added to show the patient selection, number of drop-out, and number of patients at various follow-up periods.

2. Lines 202-207: The interpretation of OR 0.77 is not sufficient and clear. Overall the objective, outcome, predictors are not clear in this paragraph. Can all the information described in this paragraph be summarized in a table to show the association between pacifier use and intervention?

3. The style of the reference should be formatted according to the PeerJ Instructions for Authors. For example, the parenthesis for the publication year should be removed, and the quotation marks for the article titles should also be removed. In addition, the volume numbers should be bolded.

Experimental design

1. Lines 50-53: The authors listed three hypotheses that can be assumed as 3 objectives in this study. However, some of the objectives are not quantitative or not clear. For example, in hypothesis 1 “infant caregivers were less familiar with the role pacifiers than…”. The outcome in this hypothesis is not clear. For example, how to evaluate the degree of familiarity quantitatively? In Hypothesis 3 “Otitis media would not occur more frequently…” Do you mean the average frequency of otitis media occurrence during the study period or just the occurrence of otitis media (i.e., yes/no)? Measurable variables should be used to describe the objectives or hypotheses.

2. The outcome measures in the methods were not described clearly. What is the primary outcome in this study? Are there any secondary outcomes? What are the definitions for the outcomes? For example, how to define the pacifier use? How to measure the knowledge of recommendations by caregivers? What are the potential predictors associated with pacifier use? How to define those predictors?

Validity of the findings

1. Data analysis: what statistical analyses were used to compare the pacifier use and other outcomes before and after the intervention? What regression analysis was performed to identify factors associated with pacifier use? What potential predictors were included in the regression model? Did you compare the pacifier use change before and after intervention by adjusting for other confounders? Did you compare the difference in occurrence of otitis media between pacifier use and none by adjusting other factors?

2. How to interpret ORs in the Table 3? Simply repeating the OR numbers is not sufficient in the results. Quantitative interpretation of the association between predictors and pacifiers using ORs is needed in the results.

Additional comments

This manuscript studied the potential of an emergency department educational intervention to increase pacifier use in infants, which is significant in decreasing the risk of sudden infant death syndrome (SIDS). However, the description about the methods used is too sketchy, more details are needed to make the methods clearer. Similarly, further interpretation is needed for the results instead of simply repeating the numbers. In conclusion, this manuscript is recommended to be published in PeerJ after major revisions.

Reviewer 2 ·

Basic reporting

There is repeat description of Table 1.

Experimental design

No comments

Validity of the findings

The authors of this paper conducted an intervention-group-only longitudinal study to test the hypothesis that an ED educational intervention would increase pacifier use in infants meanwhile the otitis media would not occur more frequently. The designs and the results are all reasonable. The issue is that only 496/780 (64%) infants were successfully followed up at three-month timepoint and 391/780(50%) were followed up at twelve-month timepoint. The losts of 36% and 50% respectively seem too high to reflect the truth of the results unless they have been proved not to affect the results. The same issue is the exclusion of the 62 infants at delayed three-month follow-up.

Additional comments

No comments

Reviewer 3 ·

Basic reporting

No Comments

Experimental design

1. Would it be appropriate to collect the data from SIDS infants to see how many were using pacifier, and compare the data with those who are using pacifier? I understand it would be of great difficulty to collect this kind of information and needs the support of those parents.

2. The prevalence of otitis media is higher of kids at two years old than those from 0-2 years old. However, the subjects included in this paper are all younger that 12 month old. What's the rational to correlate the use of pacifier to the incidence of otitis media?

Validity of the findings

Need to clarify the rational of Experiment design especially on the otitis media part.

·

Basic reporting

The submission meets or exceeds all of the requirements in this category. The concept of the teachable moment has been justified by citing literature from other fields.

Experimental design

No comments.

Validity of the findings

While 37% of non-users became users, 38% of users became non-users. The authors indicate that it is logical to assume that their intervention did not induce non-use among users. They do not comment on why users might have converted to non-use. Although they do indicate that user to non-user conversion was more common among the older infants in the cohort, there should be a more direct statement acknowledging this trend and speculating on why this might have occurred. In the least, it could be identified as a limitation of the study.(38% of users discontinued use; the study did not inquire as to the factors contributing to this change in status.)

Additional comments

The study is well written with good justification for not using a prospective randomized study design. Outcomes are measurable, and the literature supports caregiver reporting as being accurate. The use of intention to treat was justified and makes sense for the integrity of the study. the percentage of study subjects for whom follow up was successful is acceptable, but not impressive. Clearly, many caregivers informed others about what they had learned during the intervention. The intervention is inexpensive and easy, so it is very generalizable to any health care setting anywhere in the world. I believe this makes the study a valuable contribution to the fund of knowledge in Pediatric EM.

---

## Round 0.2 · accepted · Accept

I am pleased to inform you that your paper has been accepted. Congratulations.